# Establishment of Primary Adult Skin Fibroblast Cell Lines from African Savanna Elephants (*Loxodonta africana*)

**DOI:** 10.3390/ani13142353

**Published:** 2023-07-19

**Authors:** Amèlia Jansen van Vuuren, Julie Bolcaen, Monique Engelbrecht, Willem Burger, Maryna De Kock, Marco Durante, Randall Fisher, Wilner Martínez-López, Xanthene Miles, Farzana Rahiman, Walter Tinganelli, Charlot Vandevoorde

**Affiliations:** 1Separated Sector Cyclotron (SSC) Laboratory, Radiation Biophysics Division, National Research Foundation (NRF)-iThemba Laboratories for Accelerator Based Sciences (LABS), Cape Town 7100, South Africa; a.jansenvanvuuren@gsi.de (A.J.v.V.); jbolcaen@ilabs.nrf.ac.za (J.B.); mengelbrecht@tlabs.ac.za (M.E.); rg.fisher@ilabs.nrf.ac.za (R.F.); xe.miles@ilabs.nrf.ac.za (X.M.); 2Department of Medical Biosciences (MBS), Faculty of Natural Sciences, University of the Western Cape (UWC), Cape Town 7530, South Africa; mdekock@uwc.ac.za (M.D.K.); frahiman@uwc.ac.za (F.R.); 3Biophysics Department, GSI Helmholtzzentrum für Schwerionenforschung, 64291 Darmstadt, Germany; m.durante@gsi.de (M.D.); w.tinganelli@gsi.de (W.T.); 4Dr Willem Burger Consulting, Mossel Bay 6503, South Africa; wildlifevetwillem@gmail.com; 5Institut für Physik Kondensierter Materie, Technische Universität (TU) Darmstadt, 64289 Darmstadt, Germany; 6Genetics Department and Biodosimetry Service, Instituto de Investigaciones Biológicas Clemente Estable, Montevideo 11600, Uruguay; wilnermartinezlopez@gmail.com

**Keywords:** African elephant, *Loxodonta africana*, dermal fibroblasts, primary cell line, metaphase spread, skin punch biopsy, explant culture

## Abstract

**Simple Summary:**

It is paramount to preserve the genetic material of endangered wildlife species for future generations, including that of the African savanna elephant (*Loxodonta africana*). By analyzing their genetic information, it is possible to decipher how endangered species such as elephants adapted to live on our planet and how they are related to one another and to their extinct ancestors. This information can further assist us in the protection of endangered animals and the habitats they live in. One way of doing this is through the cryopreservation of elephants’ primary cell lines. This protocol presents the successful establishment of primary dermal fibroblast cell lines using a simple explant outgrowth method, starting from a small punch biopsy sample, which is minimally invasive and allows sample collection of free-roaming elephants. The average explant outgrowth, doubling time and further characterization of the dermal fibroblast cells of different elephants have been evaluated and compared. This preliminary method illustrates the potential to establish cell lines from living elephants, which contributes to the conservation of their genetic material and allows further research on the cancer suppressing ability of elephants.

**Abstract:**

Following population declines of the African savanna elephant (*Loxodonta africana*) across the African continent, the establishment of primary cell lines of endangered wildlife species is paramount for the preservation of their genetic resources. In addition, it allows molecular and functional studies on the cancer suppression mechanisms of elephants, which have previously been linked to a redundancy of tumor suppressor gene *TP53*. This methodology describes the establishment of primary elephant dermal fibroblast (EDF) cell lines from skin punch biopsy samples (diameter: ±4 mm) of African savanna elephants (*n* = 4, 14–35 years). The applied tissue collection technique is minimally invasive and paves the way for future remote biopsy darting. On average, the first explant outgrowth was observed after 15.75 ± 6.30 days. The average doubling time (Td) was 93.02 ± 16.94 h and 52.39 ± 0.46 h at passage 1 and 4, respectively. Metaphase spreads confirmed the diploid number of 56 chromosomes. The successful establishment of EDF cell lines allows for future elephant cell characterization studies and for research on the cancer resistance mechanisms of elephants, which can be harnessed for human cancer prevention and treatment and contributes to the conservation of their genetic material.

## 1. Introduction

The elephant (*Elephantidae*) population continues to decline across the African continent due to poaching, disease susceptibility, habitat loss and threats related to climate change of free-roaming elephants [1]. As a result, all species of elephants are currently listed as critically endangered, with the African forest elephant (*Loxodonta cyclotis*) critically endangered enough to be on the International Union for Conservation of Nature (IUCN) red list of threatened species™ [2,3,4]. Cryopreserved biomaterial and biobanks may offer a promising value to the conservation of the genetic material of elephants for future research and conservation efforts [5,6,7,8]. Projects such as the ‘Frozen Zoo’ and ‘Frozen Ark’ have been launched internationally with the goal to safeguard the genetic resources of threatened, rare, and elusive animal species for the next generations [9]. Hence, the establishment of primary elephant dermal fibroblast (EDF) cell lines would allow for the acquisition of large amounts of cells with most of their original characteristics preserved for functional and genetic analyses [10]. Primary fibroblast cell lines have previously been established from several endangered animal species in captivity such as collared peccaries (*Pecari tajacu*) [11], jaguars (*Panthera onca*) [12], and a wild corsac fox (*Vulpes corsac*) [13]. They have also been established from fresh carcasses of a common hippopotamus calf (*Hippopotamus amphibius*) [14], a Chinese muntjac (*Muntiacus reevesi*) [15], a Sumatran rhinoceros (*Dicerorhinus sumatrensis*) [16] and from Asian elephants (*Elephas maximus*) [17], to name a few examples. The establishment of these somatic cell lines are considered to be a valuable resource for modern somatic cell nucleus transfer as part of biodiversity conservation [5,18,19]. 

In addition to wildlife conservation, a primary elephant cell line is also highly of interest to the cancer research community because elephants appear to have developed cancer suppression mechanisms over the course of evolution [20,21,22,23,24]. The African elephant is the largest land mammal, with a body mass up to 7000 kg and longevity of up to 60 years in the wild. A simplified multi-stage model of carcinogenesis is based on the old observations that the age-specific incidence of cancer was consistent with the stepwise accumulation of 6 to 7 mutational ‘hits’ within a cell [25,26]. However, if cancers are initiated by a series of somatic mutations, species with a larger body size (more cells) and longer lifespans (more cell divisions) should have a cancer incidence that is orders of magnitude greater than small, short-lived species [23,27]. The observation that cancer incidence in large, long-living mammals, such as elephants and whales, does not correlate with species body size and longevity was first described in 1975 by epidemiologist and statistician Richard Peto and is known as Peto’s Paradox [28]. This suggests that elephants have evolved enhanced cancer suppression mechanisms in order to offset the trade-off associated with their large bodies, long lifespan and cancer incidence.

Several recent studies have investigated the genetic mechanisms responsible for cancer suppression in elephants, leading to the discovery of 19 extra copies of tumor suppressor genes *TP53* in the form of retrogenes (RTGs) and 11 extra copies of leukemia inhibitory factor (*LIF*) [20,21,29,30]. *TP53* is a particularly well-known tumor suppressor gene, mutated in approximately 50% of human cancers, and often referred to as the “guardian of the genome” [30,31]. This suggests that the cancer defense of elephants is mediated by an enhanced apoptotic response to DNA damage, in order to kill potentially cancerous cells at an early stage [23,31,32]. However, this enhanced apoptotic response in elephants needs to be balanced by other adaptations to prevent stem cell exhaustion due to the increasing need for cell replacement, which could in turn lead to premature aging, as confirmed in a study on mutant mice with increased p53 protein activity [33]. 

Several studies have used elephant fibroblast cells from the San Diego Zoo to study the cancer suppression mechanisms in elephants [21,34,35]. To the best of our knowledge, no detailed protocol on the establishment of these fibroblast cells from captive zoo elephants has been published thus far. However, one previous protocol is currently available from Siengdee et al., wherein primary fibroblast cell lines were established from post-mortem elephant ear samples [17]. In contrast to this previous protocol, our method starts from a small skin punch biopsy sample of living elephants, which allows remote biopsy darting, a technique which is no longer limited to post-mortem collection and could eliminate the need for sedation [36]. It also provides the opportunity to collect skin tissue samples from a variety of free-ranging terrestrial animals, including the elusive African forest elephant. 

This methodology describes the collection, establishment and cryopreservation of a primary EDF cell line of African savanna elephants (*Loxodonta africana*) and a method to obtain metaphase spreads from the primary EDF cultures. The successful establishment of these primary EDF cell lines allows for future studies in a broad range of research fields, including biomimetic, conservation and evolutionary biology, as well as oncology. 

## 2. Materials and Methods

### 2.1. Tissue Sample Collection

Samples of 5 elephants aged between 14–35 years old were collected at Botlierskop Game Reserve and Sanbona Wildlife Reserve in the Western Cape, South Africa (Table 1). Ethical approval was obtained from the Animal Research Ethics Committee (AREC) of the University of the Western Cape (UWC), South Africa (AR21/6/4) as was a landowner permission letter prior to sample collection. Samples were obtained from elephants who underwent a scheduled veterinary intervention (vaccination or replacement of tracking collar) requiring sedation via a single intramuscular injection of Thianil^®^ (Appendix A). The sample collection piggybacked on these interventions, which limited unnecessary sedation risks to the elephants so that no animals were sedated for the sole purpose of this pilot study. 

The skin on the backside of the elephant ear was cleaned with 70% ethanol and surgically draped (Figure 1A–C). Approximately 5–6 skin punch biopsies of 4 mm in diameter were collected with a standard punch biopsy needle behind the ear, where the skin is the thinnest [37]. The biopsies were transferred to a sterile 15 mL conical tube containing 10 mL transport media to prevent dehydration and contamination. This contained 90% Eagle’s Minimum Essential Medium (EMEM) and 10% Penicillin/Streptomycin/Amphotericin B (P/S/A) (Figure 1D). After the sample collection and veterinarian intervention, a reversal agent for the anesthetic was administered. Elephants emerged from anesthesia within 5 min after the injection and were closely monitored to ensure they reunited with the herd. Samples were transported at 0–4 °C with indirect contact to ice to prevent freezing of the biopsy fragments, to the National Research Foundation (NRF) iThemba Laboratory for Accelerator-Based Sciences (LABS) in Faure, South Africa. Upon arrival, the skin biopsies samples were processed under sterile conditions within 24 h. 

### 2.2. Biopsy Preparation and Establishment of the Primary EDF Cell Culture

All cell culture consumables and reagents (Appendix A) were sterilized and all the reagents were pre-warmed in a water bath (37 °C) before the start of the procedure. The skin biopsy was removed from the 15 mL conical tube with sterile tweezers and washed thrice with washing phosphate buffered saline (PBS), containing 10% P/S/A, and 2% gentamicin (Figure 1E) and placed in a glass Petri dish. Two scalpels (size 11) were used to remove the epidermis (Figure 1F, blue arrow) and the remaining biopsy fragment was cut into smaller tissue fragments (Figure 1F, green arrow). During manual fragmentation of the tissue, a few drops of initial culturing media (EMEM supplemented with 20% fetal bovine serum (FBS) and 1% P/S/A) were added to prevent dehydration of the tissue fragments. The small tissue fragments were transferred to a T25 tissue culture treated flask using a sterile glass Pasteur pipet (Figure 1G,H) and 0.5 mL FBS was added to the T25 flask. The T25 flasks were incubated under standard conditions at 37 °C in a humidified 5% CO_2_ incubator. After 24 h, the flasks were placed upside down for the next 24 h to improve attachment and to stop the tissue fragments from floating in the FBS. On day 3, 1 mL initial culturing media was added. The tissue fragments were monitored daily with an inverted microscope (Zeiss Primovert, Carl Zeiss Pty LTD, Munich, Germany) at 10× magnification to observe explant dislodging and the overall radial migration of the primary EDFs. Microbial or fungal contamination was closely monitored at 40× magnification. If contamination was suspected, the flask was isolated and washed daily with 2 mL washing PBS and the antibiotics of the initial culturing media was increased to a maximum of 5% P/S/A. Additionally, a commercial 16S ribosomal RNA gene polymerase chain reaction (PCR) assay was performed, and no bacterial DNA (including Mycoplasma species) was detected in the EDF cell lines.

Once a T25 flask reached 70% confluency or when over-confluent localized cell densities were observed along tissue fragments, the EDFs were washed twice with PBS, followed by 1 mL of 1× trypsin EDTA solution (0.12% trypsin, 0.02% EDTA) and incubation for 5 min. Upon detachment, 4 mL cEMEM (20% FBS, 1% P/S) was added and the cell suspension was transferred to a 15 mL conical tube. To ensure recovery of >95% of the cells, the trypsin step was repeated. After 8 min centrifugation at 201 relative centrifugal force (RCF) the pellet was gently resuspended in 2 mL cEMEM and transferred to a new T25 flask. A confluent T25 flask containing approximately 5 × 10^5^ cells/T25 flask was sub-cultured at a seeding density of approximately 3500–7000 cells per cm^2^. E1 was cultured for 12 weeks and closely monitored for morphological changes and cellular senescence. Potential morphological changes were closely monitored throughout passages with a Zeiss Primovert, light microscope and a live cell imaging system CytoSmart, Lonza^®^ (CytoSmart Technologies B.V., Eindhoven, North Brabant, The Netherlands). 

### 2.3. Primary EDF Growth Curve and Doubling Time (Td) Determination

An initial growth curve was determined for E3–E5 at passage 1. Post seeding, the cells were incubated for 2 days and from day 3 onwards, 6 images (field of view 2.40 × 1.40 mm) were taken daily in 6 different regions of the T25 flask until a confluent monolayer was reached using the live cell imaging system (CytoSmart, Lonza^®^). The confluency was defined using the CytoSmart confluency algorithm [38]. The percentage confluency (*y*-axis) was plotted in function of the number of days in culture (*x*-axis) by the means of standardization as shown below:(1)$$\frac{y}{maxiumum\;confluency\;}\times 100$$

The doubling time (Td) was determined for E3 and E5 at passage 1 and 4 by using the data points of the 6 regions in the T25 flasks and the CytoSmart confluency algorithm, using the following formula, where t represents time:(2)$$Doubling\;Time\;\left(Td\right)=\frac{t2-t1\;}{x}$$
(3)$$x=\frac{log\log\left(\frac{end\;cell\;coverage}{start\;cell\;coverage}\right)}{log\log\left(2\right)}$$

### 2.4. Primary EDF Cryopreservation and Thawing

After sub-culturing and centrifugation (Section 2.2), the cell pellet was resuspended in 90% FBS and 10% dimethyl sulfoxide (DMSO) freezing media in a cryovial. Immediately followed by transfer to a Mr. Frosty™ freezing container containing isopropyl alcohol acclimated to −20 °C overnight. Thereafter, the cryovials were transferred to −80 °C or liquid nitrogen for long term storage.

To resuscitate the EDFs, the cryovials were thawed quickly by placing them in a warm water bath (37 °C) until a small frozen clump remained and the cell suspension was added to 6 mL prewarmed cEMEM. After centrifugation at 201 RCF for 8 min, the supernatant was discarded, and the pellet was resuspended in 2 mL cEMEM and transferred to a T25 flask. Cell attachment was observed after approximately 4 h and the media was replenished every third day.

A trypan blue cell count was performed to determine the percentage of viable cells for E3 and E5 at passage 3 under three conditions: (1) prior to freezing, (2) after 7 days of freezing at −80 °C, and (3) after 14 days of freezing at −80 °C. A 1:1 ratio of trypan blue and cell suspension was used in a hemocytometer counting chamber (Marienfeld Laboratory Glassware, Baden-Württemberg, Germany). All 4 quadrants of the hemocytometer were counted for live and dead cells. The percentage viable cells in culture were calculated as follows:(4)$$Percentage\;\left(\%\right)\;viable\;cells=\frac{Live\;}{Total\;\left(Live+Dead\right)}\times 100$$

### 2.5. Metaphase Spreads of the EDFs

For the optimization of the metaphase chromosome spread presented here, the EDFs of E1 were used. Once a confluent monolayer was obtained, the cells were sub-cultured and 35,000 cells were seeded into a Petri dish (35 mm). After 90% confluency was reached, 0.05 µg/mL of KaryoMAX™ Colcemid™ Solution in PBS (stock solution 10 µg/mL) was added for 3 h. Colcemid is a mitotic inhibitor that binds to the protein tubulin and prevents spindle fiber formation and therefore causes metaphase arrest [39]. Cells were harvested by trypsinization (Section 2.2), transferred to a conical 15 mL tube and 5 mL prewarmed hypotonic solution (0.075 M potassium chloride) was added in a dropwise manner while slowly stirring and left for 10 min at 37 °C. The cells were fixed in methanol and acetic acid (3:1 ratio, freshly prepared at room temperature) using a Pasteur pipet, by slowly adding 1.5 mL fixative from the bottom to the top of the 15 mL conical tube before centrifugation for 10 min at 201 RCF. The supernatant was discarded, and this fixation step was repeated twice (5 mL fixative). The cell pellet was resuspended in 200 µL of the left over fixative and 50 µL of the cell suspension was dropped from 30 cm height on wet slides (stored in dH_2_O with 10% methanol, at 4 °C). Slides were left to dry overnight, followed by staining for 3 min with Giemsa’s Azur Eosin Methylene Blue solution in HEPES buffer working solution (1:25 ratio, freshly prepared and protected from light). The slides were rinsed In dH_2_O trice and embedded with coverslips using a drop of Dibutylphthalate Polystyrene Xylene (DPX) mounting media, once the slides were dry. The slides with metaphase chromosome spreads were initially detected using a 10× objective with the metaphase finder algorithm of Metafer (MetaSystems, Heidelberg, Germany) and subsequently captured with a 63× objective using immersion oil.

### 2.6. Quantification and Statistical Analysis

The average and corresponding standard deviations (SD) were calculated from the results of the individual experiments. Results were visualized using Microsoft Office Excel 2019 (Microsoft Corporation, Washington, DC, USA) and GraphPad Prism Software Version 5.01 for Windows (GraphPad Software, San Diego, CA, USA). As a result of limited availability and opportunistic sample collection of free-ranging elephants, not all the experiments were performed on the same day for the explant outgrowths of all 4 individuals.

## 3. Results

### 3.1. Primary EDF Outgrowth and Culturing 

The first outgrowth of fibroblast cells around the explant pieces was visible as early as day 9 (Figure 2A), but there was a large intra- and inter-elephant variation. The explant cultures of E1, E3, E4 and E5 showed the first fibroblast outgrowth at 13.25 ± 3.40, 19 ± 1.20, 25 ± 0 and 11 ± 0 days (average ± SD), respectively (see Table 2). No fibroblast outgrowth was observed in any of the E2 biopsy samples and culture flasks were discarded 40 days post the initial culturing date. Overall, the average outgrowth of fibroblasts was 15.75 ± 6.30 days (*n* = 4). 

In total, 4 out of the 5 elephants exhibited outgrowth, resulting in an overall success rate of 80.00%. The outgrowth of cells from the explant source initially resulted in a dense area with high cell numbers, which gradually migrated and spread outward over time as shown in Figure 2(A1–A3). However, not all skin punch biopsies of the same elephant resulted in successful cultures and the underlying reason for this intra-elephant variability remains unclear and requires further investigation. While 4 out of the 5 biopsies presented outgrowth for E1, only 2 out of the 6 biopsies for E4, E3 and E5 were successful (see Table 2). 

Fibroblasts are elongated spindle- or star-like cells, with cytoplasmic projections that can be short and wide or long, thin and branched. They have an elongated nucleus, limited cytoplasm and the fibroblasts grow aligned and in bundles when confluent [40,41]. So far, no maximum number of passages for primary EDF cell lines has been determined and its determination will be part of future cell characterization studies. As presented in Figure 2(B1,B2), the morphology of E1 from passage 3–14 appears to be normal throughout culturing with no clear signs of cellular senescence. 

### 3.2. Primary EDF Growth Curve and Doubling Time (Td)

The confluency of the cultures of E3, E4 and E5 was determined daily with the live cell imaging system over 11 consecutive days at passage 1. The fibroblasts of each elephant had their own replication rate and the T25 flasks reached confluency (±95%) on day 12, 17 and 11 for E3, E4 and E5, respectively. As expected, the confluency of the fibroblasts of all flasks increased gradually over time and was on average 76.45% ± 26.56% on day 11, as presented in Figure 3.

Based on the confluency algorithm of the live cell imaging system that was used to establish the growth curve presented in Figure 3, Tds of 82.15 h, 112.53 h and 84.37 h were calculated for E3, E4 and E5, respectively. This resulted in an average Td of 93.02 ± 16.94 h for the 3 elephants.

At passage 4, E3 and E5 had Tds of 52.71 h and 52.06 h, respectively. This indicates that the Td decreased in the consecutive initial passages, to an average of 52.39 ± 0.46 h.

### 3.3. Primary EDF Cryopreservation 

Prior to freezing, the percentages of viable cells of E3 and E5 were 72.05 ± 1.72% and 96.43 ± 7.14%, respectively. After 7 days of freezing at −80 °C the cells were thawed and the percentages of viable cells of E3 and E5 were 86.67 ± 9.03% and 80.42 ± 3.94%, respectively. After 14 days of freezing, the procedure was repeated and the percentages of viable cells were 79.86 ± 4.17% and 89.05 ± 10.46% for E3 and E5, respectively (Figure 4). 

### 3.4. Metaphase Spreads of the EDFs

EDFs of E1 were used to optimize a protocol to establish metaphase spreads. Once the cell density reached 90% confluency, cultures at passage 6 were exposed to 0.05 µg/mL colcemid for 3 h, which yielded metaphases with scattered chromosomes from single fibroblastic cells (Figure 5A,B). Our metaphase spreads confirmed that the diploid number of chromosomes in *L. africana* is 56, which is in accordance with the Elephantidae family and with previous publications [42,43]. 

## 4. Discussion

This methodology illustrates the successful establishment of four primary EDF cell lines from a small punch biopsy sample, following a relatively simple and cost-effective protocol, which resulted in satisfying primary dermal fibroblast cell yields and viability. This methodology can be used for future cytogenetics and conservation studies, as well as to study the mechanism of action of the *TP53* RTGs in elephants. These RTGs contain various deletions, but some copies appear to have low levels of transcriptional activity [20]. Hence, they might play a crucial regulatory role in balancing the tightly connected biological processes of cancer and aging in elephants [20,22,31]. African elephant fibroblasts have been used in the molecular analysis of *TP53*, but also in telomere length studies and for the development of elephant-specific plasmid encoding viral transforming proteins [20,21,34] Interestingly, in the latter studies, the African elephant fibroblasts were mostly sourced from the San Diego Frozen Zoo, which established the cell line from biopsies from the inner thigh at necropsy [20,21,34,35]. 

A detailed protocol on the establishment and cryopreservation of the primary fibroblast cell line obtained from the San Diego Zoo elephants has not been published, but there is a previously published methodology available on the establishment of primary fibroblast cultures from Asian elephants in Thailand [17]. In this protocol, larger ear skin tissue samples (3 × 3 cm^2^ pieces) from elephant carcasses were used for the establishment of the primary fibroblast cultures. In contrast, our protocol used much smaller skin punch biopsy samples (4 mm diameter) from living, free-roaming African savanna elephants. Our biopsy technique allows remote biopsy darting in the future, a technique that can be used to collect skin tissue samples from a variety of free-ranging terrestrial animals, including the free-ranging elusive African forest and savanna elephants without the need for sedation, bringing minimal harm to the animal. The success rate and waiting period before the first outgrowth of fibroblasts was, however, comparable between our protocol (80.00%, 15.75 ± 6.30 days) and the protocol of Siendgee et al. based on post-mortem ear tissue samples (83.30%, 4–12 days) [17]. A study on primary human fibroblasts has shown that tissue pieces with ragged edges contributed to poor attachment and cell outgrowth [40]. Another study on fibroblast cultures from mouse ear and tail tissues concluded that insufficient cutting or digestion resulted in low fibroblast recovery [44]. While it is still unclear as to why no outgrowth was observed for elephant E2 in our study after 40 days, the above-mentioned reasons could potentially play a role.

Furthermore, the variation in days upon first EDF outgrowth between the five elephants could be caused by environmental factors, such as the weather conditions when samples were obtained and the duration and conditions of the transportation. In this protocol, the aim was to keep transport duration under 10 h (depending on the location of the elephant in the game reserve) and a relatively constant temperature of <4 °C during transport with no direct ice contact. However, others have documented no effect on the cell viability upon a shipment of skin samples on ice for several days [45,46,47], so there could be other factors at play, such as the depth and composition of the punch biopsy sample itself.

In this methodology study, we opted for a direct tissue explant culture instead of enzymatic digestion. Collagenase digestion has been successful for the establishment of primary cell cultures of the Asian elephant, Jaguar (*Panthera onca*) and non-human primates from skin biopsies [10,17,33]. A protocol using dispase II, a proteolytic enzyme, followed by a simple incubation with trypsin solution has also been optimized for obtaining primary fibroblasts from living, wild berian hare (*Lepus granatensis*) and wild rabbit (*Oryctolagus cuniculus algirus*) [48]. Khan and Gasser added a collagenase D-pronase solution to their biopsies and had mouse fibroblasts adhering to tissue culture plastic surfaces within three days of culture [44]. Meanwhile, Vangipuram et al. achieved success by coating the surface of the wells with 0.1% gelatin prior to adding the skin biopsy fragment for their human fibroblast biopsies [40]. Despite a longer waiting period to establish a primary EDF culture, direct explant culture offers the advantages of cell integrity conservation and the provision of essential growth factors [14,49]. However, based on the large intra- and inter-elephant variability in successful fibroblast outgrowth, it would be interesting to compare our current protocol with enzymatic digestion methods in future studies.

The Td of the primary EDF cultures in this paper at passage 1 of 93.02 ± 16.94 h and that at passage 4 of 52.39 ± 0.46 h was significantly longer compared with the previously established hippopotamus primary fibroblast culture, with a Td of 34 h. and the Asian elephant fibroblast culture, with a Td of 25 h [14,17]. It is unlikely that age plays a role, as the median age of the elephants in this study was 25 years (age range of 14–36 years) which is much younger than the Asian elephants in the protocol of Siengdee et al. where samples were collected post-mortem (age range of 33–68 years) [17]. In the current study, no correlation could be observed between the age of the elephants, the outgrowth of fibroblasts or the proliferation rate of the fibroblasts. However, inter-elephant variations in Td and outgrowth are expected for skin punch biopsies originating from different elephants, and this was also clearly observed in the current study. E4 showed fibroblast outgrowth after 25 ± 0 days, while this was earlier for E3 and E5 (19 ± 1.20 and 11 ± 0 days, respectively). This could explain the significantly longer Td of E4 at passage 1 (112.53 h). Evidently, the passage number could influence the replication rate of the primary EDFs. The EDFs at passage 4 had, on average, a faster Td (52.39 ± 0.46 h) compared with passage 1 (93.02 ± 16.94 h). Lastly, Seluanov et al. described a Td of approximately seven days for long-lived small rodents (e.g., naked mole rats) possessing cancer suppression mechanisms, whereas human fibroblast cells divide approximately every two days [50]. The relatively slow Td of the primary EDF cultures could potentially be linked to their underlying cancer suppression mechanism, including a lower somatic mutation rate and basal metabolic rate [25].

In addition, skin punch biopsies were obtained from elephants in their natural habitat in this study, which is challenging to control and might increase the risk for microbial contamination. Therefore, a high concentration of 10% P/S/A antibiotics was applied in the transport media, in the media upon starting the cultures (1% P/S/A) and antibiotic PBS washes (10% P/S/A and 2% Gentamicin). This could potentially affect the replication rate and hence, the 1% P/S/A treatment was limited to passage 0 in this protocol [17,51]. Anti-bacterial and anti-fungal in cell culture, even standardly used P/S, can affect cell cycle regulation, differentiation, and growth [48]. Gentamicin treatment can result in mitochondrial dysfunction and oxidative damage in mammalian cells, while Amphotericin B has been shown to cause impaired physiological processes causing a decrease in cell viability [45,49]. An important foundation for genetic species conservation is the possibility to cryopreserve the EDF cells, particularly because primary cell lines are also prone to the development of senescence [11,46,47]. During the process of cryopreservation, cells must traverse a lethality zone of temperature (−15 to −60 °C) when being frozen down to very low temperatures and then once again when being thawed. In addition, while passing through this zone, cells must endure the damaging effects of vitrification, cold shock, osmotic injury, and intracellular ice formation [50]. In this current protocol, cryopreservation with 90% FBS and 10% DMSO did not induce significant effects on the cell viability of the EDFs upon thawing with an average cell viability of 84 ± 6.9% after 7 and 14 days of freezing at −80 °C. Similar conclusions have been arrived at by Borges et al., where cryopreservation did not affect the viability of their collared peccary skin-derived fibroblasts cell line when following similar freezing steps to those in our current protocol [11]. The study of Siengdee et al. showed that the Asian elephant fibroblasts had viability and post-freezing recovery rates of around 97.3 ± 4.3% and 95.5 ± 7.3%, respectively [17], which is slightly higher in comparison to our result. In the protocols of both Borges et al. and Siengdee et al., Dulbecco’s modified Eagle’s medium (DMEM) was also used in their freezing media composition, while for the current protocol no DMEM or EMEM media was used in the freezing solution [11,17]. 

Our metaphase spreads confirm the diploid number of 56 chromosomes in the *L. africana*, which is in accordance with the Elephantidae family and previous publications [42,43,52,53]. Fronicke et al. reported a predominantly acrocentric karyotype consisting of 2*n* = 56 chromosomes for the African savanna elephant containing only one subtelocentric and two metacentric autosomes. Palkopoulou et al. formally reported the high-quality reference genome of the African savanna elephant which first became available in 2005 (LoxAfr1) and has been updated in 2014 (LoxAfr4) [54]. However, it is important to mention that this investigation was only performed on the EDFs of E1 in this study, which is a limitation and additional karyotyping analysis should be performed on the EDF cultures of the different individuals in future experiments. Investigations on the ancestral karyotype of humans and elephants concluded that human painting probes produced identical hybridization patterns in both African and Asian elephants, confirming that both species possess karyotypes that differ only in the amount and distribution of C-band positive heterochromatin [44]. This type of information provides a starting point to address fundamental questions in mammalian genome architecture, including possible correlation with chromosomal polymorphisms or translocations causing neoplasia in humans. The cell lines of different elephants that were generated as part of this study might further contribute to these types of investigation [55,56].

A potential limitation of establishing a primary cell culture is the development of senescence after sequential passaging, which can vary depending on the tissue of origin and the age of the tissue source [42]. After continuous sub-culturing over a period of 12 weeks, the morphology of the EDFs was monitored closely from passage 3 to 14 (Figure 2(B1,B2)). No significant changes in the cellular size and shape, which could point to senescence, were observed. Figure 2A,B clearly confirm the presence of fibroblasts in the EDF culture based on the typical morphology displayed, which is similar to other reports on primary skin dermal fibroblasts that have reported a long, thin and multipolar appearance with disc-like structures [14,15,43]. In addition, the EDF cultures maintained an exponential proliferation rate and the known growth properties of fibroblasts. Fibroblasts can also be characterized by the expression of mesenchymal cellular markers such as vimentin and type I collagen, but this was out of the scope of the current methodology paper and will be included in future experiments [44]. While EMEM was used in this study, Siengdee et al. and Vangipuram et al. used DMEM. However, both types of media support the growth of fibroblasts whereas other cell populations (e.g., keratinocytes) need additional supplements and growth factors [17,40]. The use of additional growth factors could be an interesting approach in future studies to optimize the EDF culture conditions. Several suppliers of primary human dermal fibroblast cell lines advise the supplementation of basal media with basic fibroblast growth factors (FGFb) that signal through FGF receptors (FGFRs) and regulate biological functions, such as cellular proliferation and migration, as well as epidermal growth factor and transforming growth factor beta 1 (TGF-β1) [57,58]. Additional supplements have recently been tested by Strand et al. on their fresh or cryopreserved fibroblasts of great crested newt (*Triturus cristatus*). Three media compositions, which include insulin, mercaptoethanol and a combination of insulin/transferrin/selenite either alone or in combination with one another, were evaluated [59]. Methods to achieve cellular immortalization could facilitate future applications but are out of the scope of the current study and could potentially interfere with future studies to unravel the cancer suppression mechanisms of elephants [14,51].

This is the first experiment of its kind in South Africa, in which primary EDF cultures of four African savanna elephants were successfully established. Due to the variability in biopsy outgrowth and Td, a larger sample size might be useful to accurately determine intra- and inter-elephant variations in these parameters. However, the low accessibility, ethical and administrative constraints make it challenging to obtain biopsy samples from a large set of free-roaming elephants. It was also advised by the local animal ethics committee to avoid repetitive sampling of the same elephant. In addition, biopsy samples of free-roaming elephants have to be collected at game reserves, resulting in often unpredictable environmental challenges and higher risks for sample contamination compared with samples obtained from captive zoo elephants. As part of this proof-of-concept study, we performed opportunistic sampling and collected samples from elephants who were sedated for vaccination and collar replacements. In the case of remote biopsy darting, it might be more challenging to obtain samples originating from the skin behind the ear of the elephant where the epidermis is thin, and Global Positioning System (GPS) tracking might be required to retrieve the biopsy darts throughout the game reserves. In addition, there is a risk that the dart might penetrate the skin at an oblique angle, which would make the biopsy too superficial to reach fibroblasts. These variables will require further investigation and experimentation in order to make the remote darting technique for sample collection more successful. 

## 5. Conclusions

In this methodology paper, we present a simple method for isolating and establishing a primary EDF cell line from small skin punch biopsies of free roaming, living African savanna elephants residing in South Africa. The establishment of the primary EDF cell line brings us one step closer to cryopreserving the genome of the African savanna elephant, which is considered a valuable and critical biobanking tool for future cell-based conservation approaches. In addition, it lays the foundation for future studies in biomimetics and investigation of Peto’s paradox in the Proboscidean lineage and for unravelling the biological and genetic mechanisms that lead to cancer suppression in elephants. This protocol could also be applied for the establishment of primary cell lines of other wild endangered species. Future research should focus on immortalizing the fibroblast culture to better facilitate future applications and to overcome early senescence and further characterization of the cell line, using specific fibroblast markers such as vimentin and novel techniques such as PacBio single molecular real-time (SMRT) sequencing to establish high quality genome sequences of elephants. In addition, future research efforts should try to minimize the intra- and inter-individual variability by testing different culture conditions, optimizing transportation times and methods when more remote and elusive elephant groups are sampled without a biology laboratory in close proximity and testing the remote biopsy darting with GPS tracking. 

## Figures and Tables

**Figure 1 animals-13-02353-f001:**
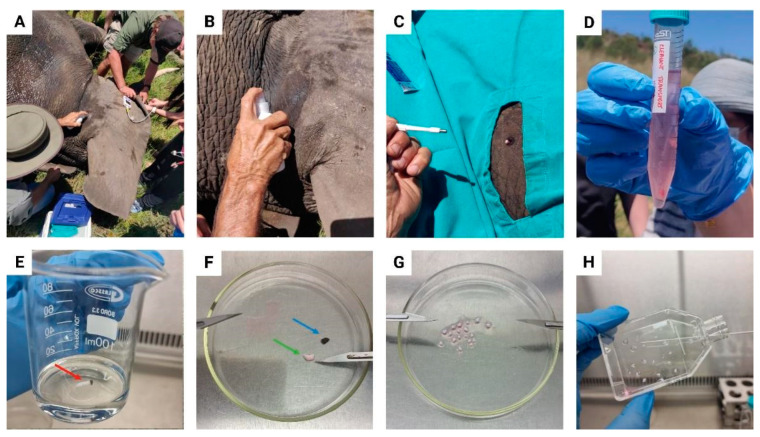
Skin punch biopsy sample collection and biopsy preparation. (**A**) Skin biopsies were collected behind the ear of the elephant. (**B**) The skin was washed and cleaned with 70% ethanol. (**C**) The punch biopsy area was surgically draped, and skin biopsies were collected with a skin punch biopsy needle. (**D**) Skin punch biopsies were added to a 15 mL conical tube containing transport media. (**E**) In the laboratory, the biopsies (indicated with a red arrow) were washed thrice with washing phosphate buffered saline (PBS) and transferred to a sterile glass Petri dish. (**F**) Scalpels were used to remove the epidermis (indicated with a blue arrow) from the biopsy fragment (indicated with a green arrow). (**G**) Initial culturing media was added to prevent dehydration of the biopsy fragment and was cut into smaller tissue fragments. (**H**) The tissue fragments were transferred to a T25 tissue culture treated flask using a sterile Pasteur pipet and 0.5 mL fetal bovine serum (FBS) was added and the T25 flask was incubated under standard culturing conditions.

**Figure 2 animals-13-02353-f002:**
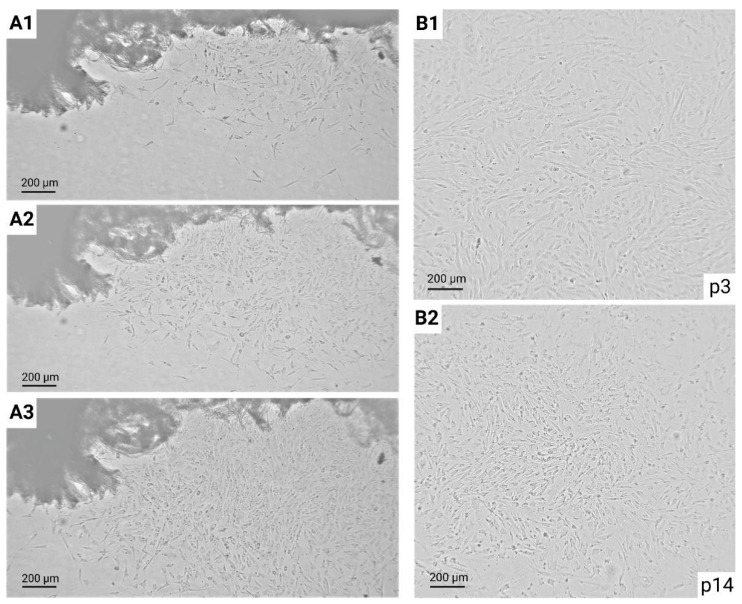
The characterized outgrowth of the primary EDFs and representative fibroblast morphology. (**A**) The radial outgrowth of EDFs from tissue fragments of E1 on day 9 (**A1**), 10 (**A2**) and 11 (**A3**). (**B**) The normal fibroblast morphology of the EDFs remained throughout passages 3 (**B1**) to passage 14 (**B2**), with elongated, disc-like structures. All data were captured with a live cell imaging system (CytoSmart, Lonza^®^) (p: passage).

**Figure 3 animals-13-02353-f003:**
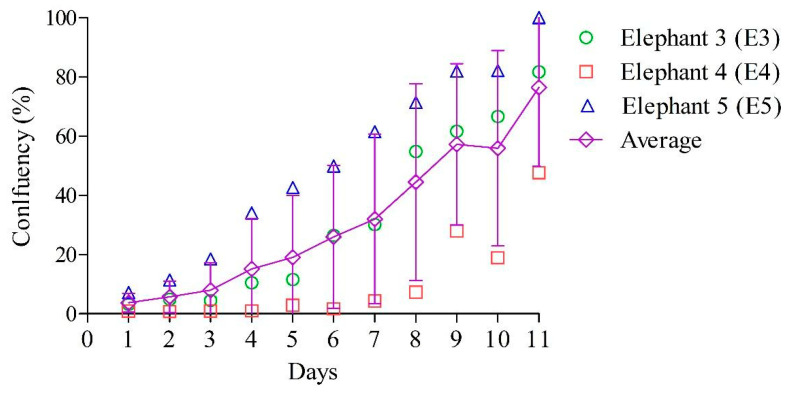
The cell confluency of E3, E4 and E5 at passage 1. Cell confluency was defined daily and the average of the 6 regions of a T25 flask was plotted graphically. On day 11, the confluency of the fibroblasts was 76.45 ± 26.56% (average ± SD). Based on the confluency algorithm of the live cell imaging system, CytoSmart, Lonza^®^, a Td of 93.02 ± 16.94 h (*n* = 3) was obtained at passage 1.

**Figure 4 animals-13-02353-f004:**
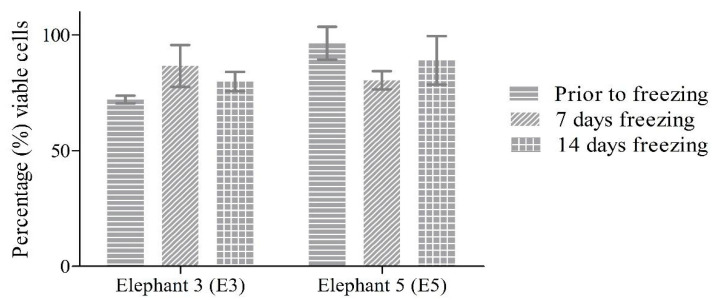
Percentages of viable cells of E3 and E5 prior to freezing and after 7 and 14 days of freezing at −80 °C. The viable cell percentages for both E3 and E5 were above 70% throughout all of the conditions.

**Figure 5 animals-13-02353-f005:**
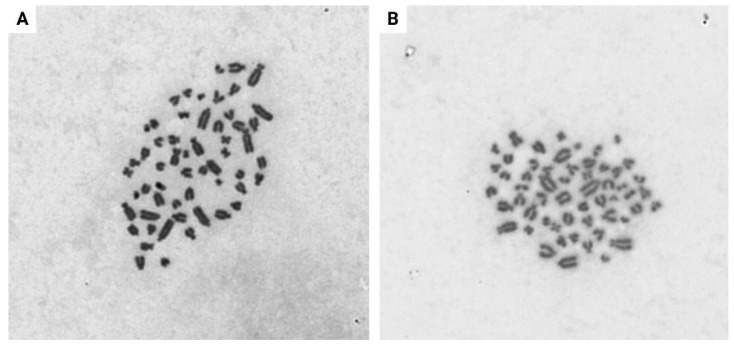
Primary EDF metaphase spread of E1 at passage 6. The images (**A**,**B**) were captured using a Metafer 4 platform. A diploid chromosome number of 56 chromosomes is visible.

**Table 1 animals-13-02353-t001:** Elephant sample collection information.

Elephant Code Name	Gender (Male/Female)	Age	Date of Sample Collection	Location of Sample Collection	Type of Analysis
E1	M	25	25.09.2021	Botlierskop Game Reserve	Morphological outgrowth; explant outgrowth; metaphase spread
E2	M	14	19.11.2021	Botlierskop Game Reserve	n/a
E3	F	35	16.12.2021	Sanbona Wildlife Reserve	Explant outgrowth; growth curve and Td *; cryopreservation
E4	M	25	16.12.2021	Sanbona Wildlife Reserve	Explant outgrowth; growth curve and Td *
E5	M	26	15.01.2022	Botlierskop Game Reserve	Explant outgrowth; growth curve and Td *; cryopreservation

Td *: Doubling time.

**Table 2 animals-13-02353-t002:** Summary of elephant sample collection and radial explant outgrowth of fibroblasts.

Elephant Code Name	Biopsy	Days for Outgrowth Confirmation	Successful Fibroblast Outgrowth/Total Number of Biopsies	Average Days until Explant Outgrowth of EDFs * (Average ± SD)
E1	1 2 3 4	16 9 - 12	4/5	13.25 ± 3.40
E2	1–5	-	0/5	n/a
E3	1 2 3 4 5	18 - 25 - -	2/6	19 ± 1.20
E4	1 2 3 4 5 6	25 - 25 - - -	2/6	25 ± 0
E5	1 2 3 4 5 6	11 - 11 - - -	2/6	11 ± 0

EDF *: Elephant dermal fibroblast.

## Data Availability

The generated and analyzed datasets which support the findings of this study are not openly available but will be made available by the corresponding author upon reasonable request.

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
