# Peer review of "Establishment of Primary Adult Skin Fibroblast Cell Lines from African Savanna Elephants (Loxodonta africana)"

_animals, 2023, doi:10.3390/ani13142353_

Round 1

Reviewer 1 Report

The authors present a manuscript describing a promising technique to collect skin biopsies and establish primary adult skin fibroblast lines. Overall, the involved procedures are described correctly and with detail, enough to be replicated by other colleagues, helping to improve the general knowledge on this subject. Therefore, the authors should receive credit for their work.

However, I do have some corrections/suggestions that I think would definitely improve this manuscript:

1) L27,33... There is no need to unnecessarily repeat "African savanna elephants" after mentioning this term for the first time with its scientific name (L23). For instance, using "elephants" is more than enough and its already implicit which species you are referring to. 

2) L66-68: since it is the first time you mention these species, you should provide its scientific name or genus/family (in those cases you are referring to more than one species). Later, you can only use the common name but the scientific name needs to be presented at this stage.

3) L101-103 and L371-373: this information (regarding the other skin biopsy technique, of 9 cm3, described by [17]) is repeated in the Introduction and Discussion. I believe it should only be mentioned in one of these sections and I would definitely choose to use it in the Discussion, to highlight the advantages of your technique and compare your results with those. There is no need in repeating information, and this is not desirable in a Technical Report (that should be objective and concise). 

4) L123-124: the chemical composition of Thianil is not mentioned here or in Supplementary table 1. I believe the chemical name of the drug combination should be provided somewhere in this paper.

5) Discussion "sections" - it seems due to the paragraph spaces that the discussion is divided into subsections, which is not a problem if authors decide to use titles for each one of them. 

6) Conclusions: authors should develop the conclusions a little bit more. Answers to the following questions can and should be provided with this purpose: What is the importance of this developed procedure under elephants' conservation? What are the limitations of its use? What could be done differently? What do you suggest for future research on the subjected (e.g. trying something differently, use it in other species ....). 

Reviewer 2 Report

The manuscript by Amelia Jansen van Vuuren et al. demonstrates very interesting and valuable study of establishment of primary skin fibroblast cell lines form African savanna elephants. Presented technical note provides information about new adult skin fibroblast cell lines that can be used in a broad range of research fields.

Critical questions which should be answered in major revision are:

1.        In Materials and Methods section – I recommend to include information about individual cultures (e.g. E1 – cultured for 12 weeks; type of analysis…; E2 – no fibroblast outgrowth ect) in the Table 1. It will help readers analyse results and discussion. 2.        In Results and Discussion section Authors  did not analyse any cellular marker (e.g. vimentin) especially in case of higher passages that provide information about identity of cell lines. Why? One of the method is karyotyping, but every cell lines should be precisely characterized. 3.        In Discussion – Did authors compared culture media composition (e.g. presence of growth factors ect) it might be helpful to analyse some differences in cellular growth of cell lines presented In this study and growth of other cellular models, please add some comment on this.  

Minor revision:

1.        In Introduction section – there are some errors in the format - the lack of spaces between words. Please carefully check and modify.

2.        In Abstract, Introduction and Discussion sections  – “TP53” as a symbol for gene should be italicized.
